# Assessing the Role of Systems Thinking for Stocker Cattle Operations

**DOI:** 10.3390/vetsci10020069

**Published:** 2023-01-18

**Authors:** Daniel B. Cummings, John T. Groves, Benjamin L. Turner

**Affiliations:** 1Boehringer Ingelheim Animal Health USA Inc., Duluth, GA 30096, USA; 2Livestock Veterinary Service, Eldon, MO 65026, USA; 3Department of Agriculture, Agribusiness, and Environmental Science and King Ranch® Institute for Ranch Management, Texas A&M University-Kingsville, Kingsville, TX 78636, USA

**Keywords:** bovine respiratory disease (BRD), systems thinking, Iceberg Diagram, creative tension, unintended consequences, archetypes

## Abstract

**Simple Summary:**

Stocker cattle operations are challenged with complex economic and environmental conditions often culminating in the increased risk of bovine respiratory disease (BRD). Innate to stocker systems is the interconnectivity of erratic intrinsic and extrinsic conditions resulting in the inability to identify viable long-term solutions to solve BRD. Animal health professionals may consider the discipline of systems thinking as an alternative approach to address the complex nature of multifactorial disease manifestations, i.e., BRD. This article provides the framework to understand the essentials of systems thinking and apply key fundamentals.

**Abstract:**

Bovine respiratory disease (BRD) is recognized as a complex multifactorial disease often resulting in significant economic losses for the stocker industry through reduced health and performance of feeder calves. Conventional approaches to manage BRD in stocker production systems can be challenged with a restricted view of the system, most importantly the structure, which drives the behavior of the system and fails to anticipate unintended consequences. The translation and implementation of systems thinking into veterinary medicine can offer an alternative method to problem-solving. Fundamental to the success of the systems thinker is the conceptualization of the Iceberg Diagram intended to identify root causes of complex problems such as BRD. Furthermore, veterinary and animal health professionals are well-positioned to serve as facilitators to establish creative tension, the positive energy necessary to identify high-leverage strategies. The interrelationships and interconnected behaviors of complex stocker systems warrant an understanding of various archetypes. Archetypes provide the systems thinker with a decision-making tool to explore tactics in a nonlinear fashion for the purpose of recognizing short- and long-term outcomes. Developing literacy in the discipline of systems thinking will further equip professionals with the skillset necessary to address the multitude of challenges ingrained in complex stocker cattle systems.

## 1. Introduction

Stocker cattle operations serve a critical role in the beef cattle supply chain. Situated between the cow-calf, feeder, and packer stages of production, the stocker segment possesses a unique set of features or characteristics. Like the cow-calf segment, stocker operations are primarily land-based enterprises, making them subject to the same climate and ecological forces that drive forage and animal productivity. Like the feeder segment, animals are procured from upstream in the supply chain and later sold downstream. This introduces tremendous financial and production risk to the stocker enterprise through input- and output-price volatility and the physical necessity of commingling and transporting animals, both of which are conducive to disease incidence and spread. As the link between cow-calf and feeder segments, the stocker segment is constrained by the temporal and spatial distribution of the supply and the quality of weaned calves throughout the year (which also varies with environmental and market adaptations to the cow-calf sector) as well as the demand for feeder cattle by feeder operators downstream.

Because of these features, stocker enterprise managers are pressed to meet their production and financial goals under complex economic and environmental conditions that are highly dynamic over time. In such circumstances, the ability of managers to correctly intuit or predict the long-term outcomes of their actions is especially weak [1,2,3]. In response, managers rely on a set of commonly accepted practices or decision strategies in order to cope with the complexity of their situation [1,4,5]. In the stocker segment, these may include feed additives, metaphylaxis, vaccinations, mineral injections, nutraceuticals, or other animal health technologies. The results of stocker management strategies not only feed back to the performance of the individual stocker enterprise but also influence the performance of feeder and packer segments given the flow of cattle through the supply chain.

The interconnectivity between segments of the beef supply chain, the economic and environmental dynamics that influence the stocker segment, and the decision-making habits that managers find themselves in given their role and position in the beef production system necessitate a systems thinking methodology to problem solving.

The systems thinking methodology was pioneered at MIT by Jay W. Forrester beginning in the 1950s [6] and has developed into a rich set of both qualitative [7] and quantitative [8] approaches to understanding and managing complex systems. The systems thinking approach has been applied to a wide variety of perplexing problems in, for example, strategic business management [9], engineering projects [10], water resource management [11,12], soil and land conservation [13], and environmental sciences [14]. The approach has also been applied to better understand animal science problems in nutrition [15], cow-calf production [16,17,18], beef cattle disease management [19,20,21], and veterinary practice [22]. These investigations and many others have shown that management efforts to address immediate symptoms of complex problems have generally been low leverage, are effective only in the short run, and often worsen problems in the long run.

To be more effective, the systems thinking process begins with reflection and dialogue about two important questions:What goal or desired outcomes are we aiming to achieve?What is our current reality (i.e., current condition or level of performance)?

The pressure that managers feel to “close the gap” between current reality and desired outcomes is called creative tension—a force that compels us to work on improving current performance to align with a goal. Although goals or desired outcomes may be shared or easily agreed on by stakeholders in a particular context, often there is disagreement about the current condition and the processes and factors that thwart management improvement efforts. The process of systems thinking provides a framework and a set of tools to better understand why problems persist in spite of our best efforts to resolve them and, therefore, more effectively reduce creative tension.

These tools begin with a conceptual exercise known as the Iceberg Diagram (Figure 1) [23]. The “tip” of the iceberg, easily seen from a surface-layer awareness of a problem, includes observations of discrete events and trends and patterns over time in key variables associated with the problem. The bottom of the iceberg, which resides far below the surface but where the majority of its mass resides, is called structure. Structure includes the biological, physical, ecological, economic, managerial, political, and industrial relationships that interact with and feed back on individual decisionmakers, their mental models, and their choices. Structure creates and gives rise to the trends, patterns, and events we see around us. The structural level of the iceberg, depicted as causal loop diagrams including archetypes, stock-and-flow simulation models, or combinations thereof [11], improves focus on root causes of issues. The structural level also helps identify and clarify testable high-leverage strategies to improve performance and close the gap between current and desired outcomes.

Some commonly shared perspectives will be articulated in the following sections regarding current and desired outcomes in the stocker industry pertaining to bovine respiratory disease (BRD) complex. Several structural-level explanations, illustrated in causal loop diagrams, for why current BRD performance in the stocker segment has stalled or eroded will be described. Actionable steps that managers or veterinary practitioners can take to better utilize the systems thinking toolbox will be discussed throughout.

## 2. Establish Creative Tension

Animal health professionals have become increasingly frustrated with stagnant or rising levels of BRD, a cause of significant economic losses in North American beef cattle populations [24]. Despite extensive research efforts to better understand this complex disease in addition to pharmaceutical interventions, improved nutrition, and preconditioning programs, the stocker industry continues to be plagued with high occurrences of BRD [25]. The challenges associated with BRD in stocker operations may result in unhealthy conflict between team members within the structure of an organization. Conflict is a natural occurrence when challenges arise and differing opinions exist. Before stakeholders, including veterinary professionals, can identify leverage points to influence the behavior of a system, they must redirect conflict to establish creative tension within the organization. Creative tension was first introduced by systems scientist and author, Peter Senge, to describe the gap between current and desired reality [26]. The following will serve as an introductory model to generate creative tension.

According to Senge, the natural tendency of tension is to create resolution by closing the gap between what is real and what is ideal [26]. Creative tension is the source of energy that fuels change to develop and implement long-term solutions while simultaneously anticipating future problems. Unfortunately, alternatives to generating creative tension are typically met with less resistance and may require fewer resources short-term. The establishment of creative tension to address BRD within a stocker operation is reliant upon the willingness of all individual stakeholders to invest and engage in dialogue.

For the purposes of this section, dialogue is defined as “sustained collective inquiry into the processes, assumptions, and certainties that structure everyday experience” [27]. The practice of dialogue is much more than debating and defending one’s views on a subject matter. Dialogue is never-ending and should consist of constructive feedback, active listening, and ongoing inquiry into the behavior of a system and its underlying structure. Well-trained professionals can be equipped with the knowledge and scientific understanding to serve as the facilitators of shared thinking and open dialogue. As a facilitator, it may be important to highlight individual contributions to the problems; however, from a systems perspective, no one individual is at fault with regard to the failure of the system. Aligning stakeholder mental models and finding agreement about the organization’s current reality and shared vision are enhanced through facilitated dialogue in a climate of openness, one which values inquiry and input from all voices as well as a collaborative effort to identify critical feedback loops at work in the organizational structure.

### 2.1. Identify Current Reality

Linear, short-term solutions to address BRD without regard to long-term feedback delays are often implemented in stocker operations with varying levels of success. Such solutions often create unintended consequences, which if not properly anticipated or recognized, can make BRD management more difficult long-term. Quick, short-term fixes to address complex problems usually create oscillating patterns with increasing variation in peaks and valleys over time (Figure 2). This behavior commonly appears as a fix in the system; however, the fix can result in backfires, which may mature over months or years. Additionally, unintended problems resulting from long-term feedback delays develop slowly and weaken managers’ ability to anticipate, account for, and respond to emergent issues. Also inherent to the system is the idea that maintaining certain levels of BRD within the individual stocker operation is an intended behavior necessary for financial gain. The business model for margin operators is to efficiently increase the value of a given commodity such as the abruptly weaned immunologically naïve calf. Knowingly, some stocker operators may accept the risk of BRD or other diseases accompanying this population of calves to realize an overall increase in returns.

The abovementioned complexities encountered in stocker systems are indicators for stocker operation stakeholders that it is time to start the dialogue needed to share individual perspectives and generate alignment about the current reality. As facilitators endeavoring to initiate this dialogue, animal health professionals may consider questions such as those listed in Table 1 to better understand the current condition of the system. In addition, accurate health and performance records should be used to identify BRD trends and patterns. Meaningful data will support the team’s collective understanding of current reality and perhaps spark the creativity needed to develop a shared vision. Aligning on the current picture of reality is a critical step to garner complete agreement on the desired outcome. For example, if caretakers do not share the same view as managers of current reality (high BRD morbidity), then it is unlikely a shared vision of the desired outcome (low BRD morbidity) will be established.

### 2.2. Develop a Shared Vision of a Desired Outcome

A shared vision for a system can be described as positive or negative according to Senge [28]. Negative visions are created based on fear whereas positive visions stem from aspiration. In this context, the default vision is “negative” for many stocker operations, i.e., avoid significant morbidity or mortality attributable to BRD. Negative visions lack innovation, promote powerlessness, and appear short-sighted. The energy for negative visions only persists when the threat is present. For example, if the threat of BRD mortality is mitigated due to external factors such as seasonal changes, then the operation’s vision and energy are lost because the threat no longer exists. The source of energy to create a positive shared vision is aspiration [28]. Aspiring to promote the health and well-being of cattle creates a sustainable, positive vision providing a continuous source of energy for learning and growth.

Many individual stakeholders and external organizations may have a functional role contributing to the complexity of stocker operations, especially with the increasing diversity currently observed in some business models. The development of a shared vision requires input from all parts of the system including nutritionists, agronomists, professional salespersons, financial providers, animal health providers, employees, regulatory agencies, government agencies, business partners, etc. Veterinary professionals are well-positioned to facilitate the dialogue necessary for all stakeholders to collaborate when attempting to improve stocker cattle health. Table 2 provides a list of potential questions to consider for stocker operators or facilitators working to align on a desired outcome.

### 2.3. Considerations When Establishing Creative Tension

Establishing creative tension is a difficult task and may be met with criticism in stocker operations due to past experiences or potential biases, such as preference to maintain the status quo. A high level of safety and trust must exist between invested team members to foster a positive environment that will allow for respectful dialogue. A lack of psychological safety and fear of failure can erode trust, producing a quiet, morbid culture. A culture in which personnel lack the confidence and security necessary to engage in crucial dialogue must be rebuilt before attempting to establish creative tension. Finally, all team members must be allowed an opportunity to provide feedback, particularly when managing complex problems such as BRD. Observations and experiences related to the management of BRD may differ depending on various responsibilities. High-leverage interventions are more likely attainable when creative tension is established through input from all components of the system.

## 3. Application of Systems Thinking for Stocker Operations

Thinking deeply about complex, persistent, and refractory problems affecting the health, productivity, and resilience of cattle production systems is a core activity of the systems thinking veterinarian or manager. For many engaged in the systems approach, this “admiring the iceberg” occupies much of the effort required to gain a deeper understanding [29]. Fundamental to effectively “admiring the iceberg” is a level of understanding and appreciation that is gained through experience and immersion within the system. This principle is particularly germane when gaining deeper understanding of the systemic forces driving BRD in production systems. A fundamental concept of a systems approach is to include perspectives and understandings beyond traditional science; it should include all system participants so that a more complete and broad knowledge regarding the “iceberg” can be gained and interconnections better understood. This concept requires that our professional community gain literacy in the discipline of systems thinking.

Often the impetus in trying to understand complex problems more deeply is attempting to recognize existing and emergent unintended consequences that are rampant in stocker systems. More importantly, pursuing an awareness of the origin or source that led to the unintended consequence in the first place is of greatest impact. For many, this thought exercise is the central and overriding activity of systems thinking. On first impression, this activity seems straightforward. However, because the fundamental causes and unintended systems behaviors are often displaced in time and space from each other, the connections are often counterintuitive and may require massive investments in time and thought to uncover. Archetypes are often used as tools for thought exercises and help systems thinkers explore and evolve their perspectives of complex problems in ways that can be helpful in overcoming mental models and breaking out of linear thought patterns.

“Fixes that Backfire” is one of the easiest archetypes to identify and appreciate and often provides a foundational understanding of systems interconnections. The central theme is that policies and decisions have both short-term and long-term consequences. The short-term fix is usually the modus operandi fix to the problem that is being addressed, which can be represented by a balancing loop in which the problem is alleviated. In the example at hand, antimicrobials have emerged over time as one of the primary tools that veterinarians and producers use to manage BRD (Figure 3). The fix can be implemented quickly, and the mitigation of the problem is nearly immediate in this balancing loop [30].

However, as is known, there are numerous ramifications associated with antimicrobial use. These ramifications are the “unintended consequences” that will occupy the systems thinker’s time when using this archetype in the thought exercise of “admiring the iceberg”. The unintended consequence under consideration is represented by a reinforcing loop known as a vicious cycle, which will worsen the initial problem that is targeted to be corrected with the short-term fix (Figure 4).

Veterinarians are often aware of the longer-term negative consequences but may choose to rely solely on the fix because the results are immediate and the need is so great. The awareness and influence of the unintended consequence loop is somewhat appeased by the delays associated with these interconnections. As antimicrobials are used to manage BRD, the reward is a nearly immediate balancing feedback loop. Meanwhile, the delays associated with reinforcing loops can be months, years, or decades, often relegating the unintended consequence to a lower priority policy issue. This delayed property of the “backfire” leads to the surreptitious return of the original problem in a condition that is incrementally worse than when the original short-term fix was applied. Often, unintended consequences are only recognized and addressed when they reach critical thresholds associated with the characteristic exponential growth of reinforcing loops. While this relatively straightforward archetype example may seem overly simplified and remedial initially, it can serve well as an introduction to a tool for systems thinking. More importantly, it may challenge some mental models that interfere with the ability to think creatively and unencumbered.

As the causal balancing and reinforcing loops of “Fixes that Backfire” are studied, they require the contemplation of several important concepts:What role are the veterinarians’ activities playing in the system? Although veterinarians devote their professional activities to mitigating and managing BRD, do they also play an important role in enabling its long-term persistence?When considering the problem being addressed with the short-term fix, is it possible that it is just a symptom of a larger system problem?

“Fixes that Backfire” can be challenging and frustrating as a framework because it focuses mostly on exploring causal relationships that are often counterintuitive and difficult to connect. It also lacks a level of thinking that allows thought about how long-term consequences and policies affect system behavior.

As the goals of systems, the causal relationships of system structure, and the mental models of people within them are more deeply understood with regard to unintended consequences, systems thinkers can utilize the “Shifting the Burden” archetype. This archetype is applied to delve into the tradeoff between short-term actions taken to address the symptoms that emerge from a complex system problem versus longer-term actions taken to address the fundamental causes of that problem that would lead to more desirable outcomes [30]. As described in the previous example, BRD is a challenge that is often seen as the central problem initially. After reflection and deeper investigations into the systems level interactions, BRD can more accurately be viewed from this perspective as being a symptom of deeper fundamental problems in the system (marketing channels that require commingling, long-hauls, poor husbandry, chronic stress, immune dysfunction, etc.) [31,32]. When viewed from this perspective, actions taken to address fundamental causes can also be considered in the framework.

The basic template for “Shifting the Burden” has two balancing loops. One balancing loop represents the short-term or symptomatic method of addressing the problem. The second loop is another balancing loop that represents the long-term effort to address the root or fundamental issue that is producing the symptom (Figure 5).

The insights from this archetype are generated when addressing symptoms of deeper challenges within the system. Addressing these symptoms can generate unintended consequences that impede our ability to focus resources long-term and ultimately delay correction of root causes. Over time, more attention and additional resources are diverted to battling symptoms instead of addressing root causes. This tendency to become dependent on symptomatic fixes rather than fundamental solutions is the reason this archetype is known as the archetype of addiction [30]. Observations by systems thinking veterinarians in the field have suggested that antimicrobials have served as an engine of growth in many feeder cattle systems, fueling operational growth and increased revenues [33]. This proposed “unintended consequence” is represented in the archetype by the addition of a vicious cycle reinforcing loop (Figure 6).

With the addition of this proposed reinforcing loop, the systems thinker is suggesting that antimicrobial use has fueled growth and revenue generation. Due to rapid growth, feeder cattle systems have realigned resources to take advantage of the relationship while simultaneously neglecting the need to make meaningful progress in correcting root causes. The thought clouds arising from different variables of the archetype describe deeper elements of industry structure or managerial mental model goals and constraints potentially contributing to “Shifting the Burden” of BRD control to antimicrobial use. Of course, the reader has the prerogative to accept or reject this hypothesis, and in fact, readers are encouraged to develop and critique as many potential “unintended consequences” as they wish. Often, a key leverage learned is the awareness of the lack of organizational effort focused on long-term and fundamental solutions. With some experience and practice, the “Shifting the Burden” archetype becomes a powerful framework to explore possible leverage points in the system. It also allows the systems thinker to powerfully communicate thoughts and concepts to others.

The application of systems thinking is not limited to addressing BRD in stocker operations. Identifying leverage points using the framework outlined above may also prove beneficial when faced with various complex animal health challenges including recurring lameness, acidosis, infectious bovine keratoconjunctivitis, coccidiosis, gastrointestinal parasitism, etc. Furthermore, as the animal health professional becomes proficient in systems thinking, this approach could be considered more broadly within the structure of the stocker operation. This idea is particularly warranted as the stocker operator’s business model becomes increasingly diverse, responding to pressures associated with both non-agricultural- and agricultural-related interests.

Specific examples of improvement in BRD-associated performance and health metrics are not reported in this article. Current research efforts to evaluate the BRD complex are often limited to response variables captured with short-term metrics associated with disease in the individual animal or group, not the health and well-being of the system. Measuring the positive or negative impact of systems thinking through applied research is challenging in an environment limited by time and resources. Veterinary practitioners, academicians, industry partners, producers, consumers, and other stakeholders must collaborate to conduct meaningful applied research to objectively measure the quantitative and qualitative benefits of harnessing the systems thinking approach to improve stocker operations management in the future.

## 4. Conclusions

Animal health professionals and managers working in stocker systems often encounter highly complex problems for which solutions are not easily identifiable. Applying interventions identified through linear thinking, while highly innovative from a scientific perspective, often fails to adequately recognize feedback and identify leverage points within the system. Linear thinking focused on short-term fixes can evolve into unintended adaptive consequences leading to more complex problems over time. In contrast, systems thinking is intended to understand why problems persist and how specific actions impact the behavior of the system. Engaging in extensive dialogue and implementing the fundamentals of systems thinking outlined in this article will afford animal health professionals the opportunity to identify leverage points to influence the behavior of the system. Strategies to improve focus on root causes, establish creative tension, and explore various mental models through archetypes are key initiatives for the systems thinker. Aspiring systems thinkers are encouraged to review additional resources related to the field of veterinary medicine to further understand and apply systems thinking when challenged with complex problems [34].

## Figures and Tables

**Figure 1 vetsci-10-00069-f001:**
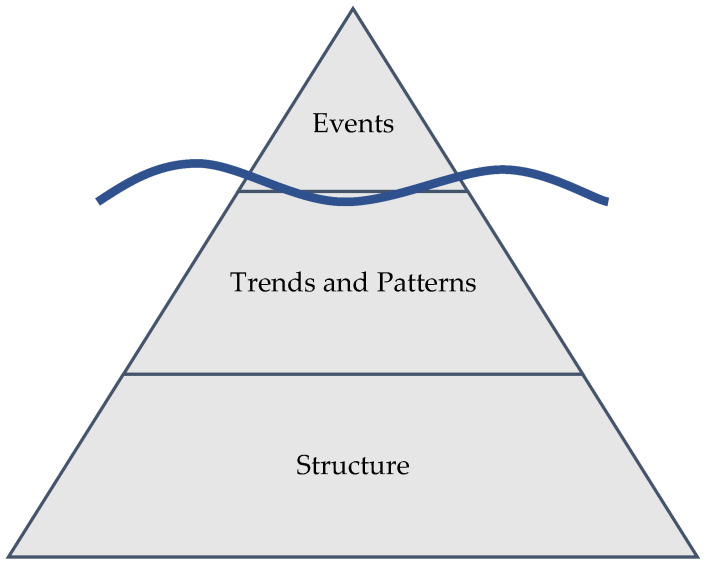
Iceberg Diagram depicting structure as the level giving rise to trends, patterns, and events.

**Figure 2 vetsci-10-00069-f002:**
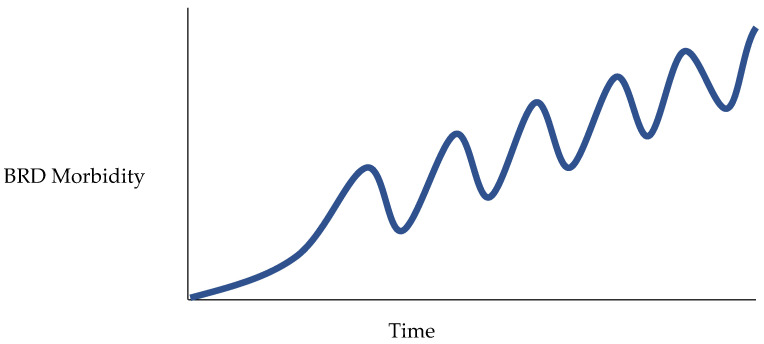
Oscillating pattern of BRD morbidity observed with short-term solutions.

**Figure 3 vetsci-10-00069-f003:**
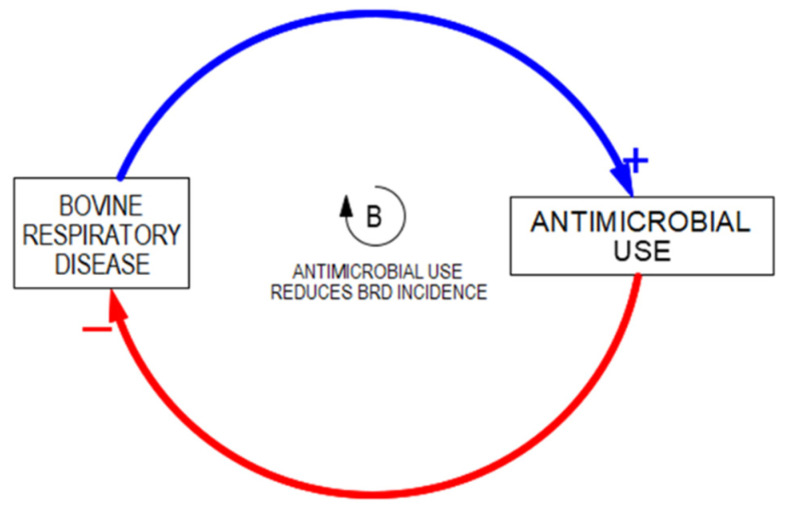
Balancing feedback loop signifying alleviation of a problem with a short-term fix. Arrows connect variables and convey the direction of the relationship with the “+” and “−” notation.

**Figure 4 vetsci-10-00069-f004:**
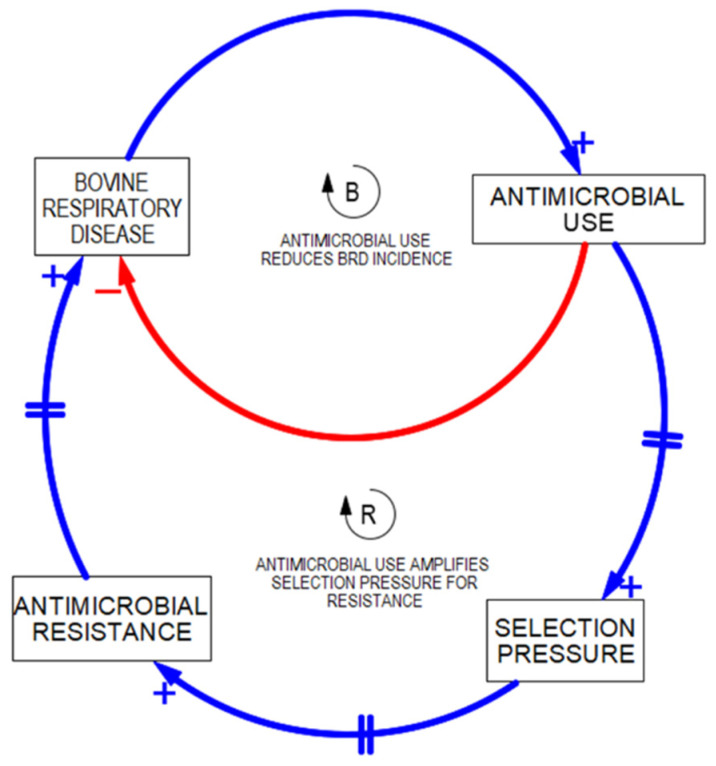
Balancing loop with reinforcing loop demonstrating an unintended consequence.

**Figure 5 vetsci-10-00069-f005:**
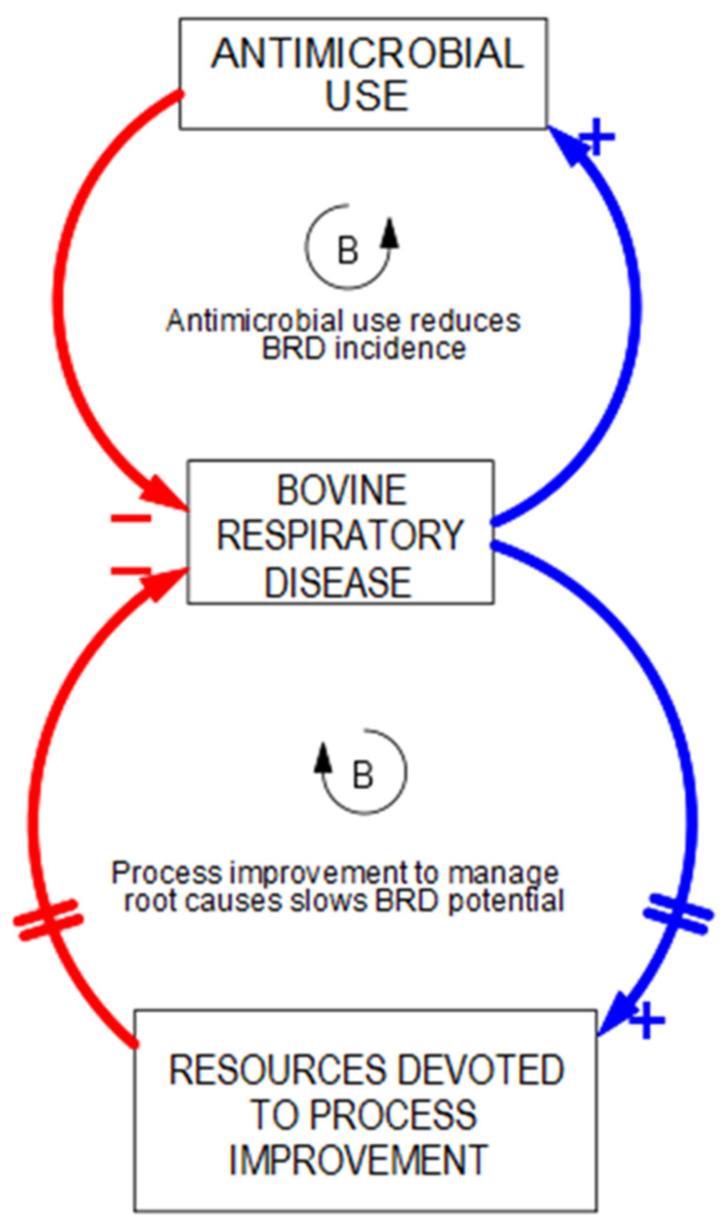
Causal loop diagram representing the basis of the archetype, “Shifting the Burden”.

**Figure 6 vetsci-10-00069-f006:**
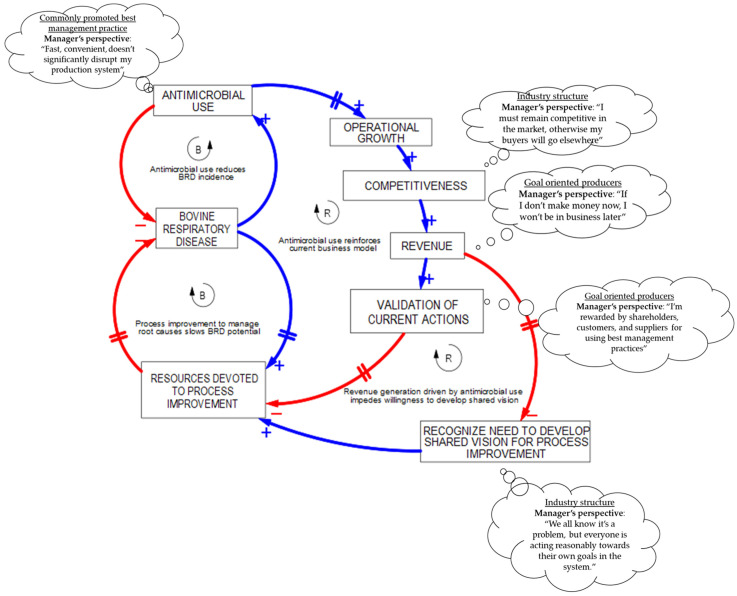
“Shifting the Burden” causal loop diagram with two reinforcing loops representing the side-effects of antimicrobial use on growth and revenue long-term and, therefore, willingness to recognize the need to develop a shared vision.

**Table 1 vetsci-10-00069-t001:** Probing questions to consider when identifying the current reality of a complex problem.

Questions to Identify Current Reality	Expand
What events are happening that capture our attention?	Why?
What do we not understand about BRD in our operation?	Why?
What are the current BRD trends?	Why?
What are the emerging health patterns?	Why?
What are team members currently facing or experiencing?	Why?
What aspects or resources from our current reality can be leveraged?	How?

**Table 2 vetsci-10-00069-t002:** Probing questions to consider when developing a shared vision of a desired reality.

Questions for Developing a Shared Vision
Who needs to be included in the process of creating a shared vision? - Have any employees or stakeholders been excluded in the past? - Will any need to be included in the future?
Ideally, what does BRD prevention and control look like for our operation?
What resources are we willing to allocate toward our efforts to control BRD?
What may be encountered 5–10 years from now because of achieving our vision?

## Data Availability

Data sharing not applicable. No new data were created or analyzed in this study.

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
