# Peer review of "Assessing the Role of Systems Thinking for Stocker Cattle Operations"

_vetsci, 2023, doi:10.3390/vetsci10020069_

Round 1
Reviewer 1 Report
well written paper introducing the systems thinking framework
it would be good to have a specific example related to BRD with actual numbers and intervention/leverage points to see the benefit of the systems thinking approach and how key fundamentals are measured e.g. mortality of 20%, short term strategy decreased to XX % and implementing the long term strategy decreased to XX%
it would be good to see links shown between system thinking frameworks for BRD, acidosis, lameness for example and how it is essential to apply the framework across the business
Author Response
- Specific example of improving BRD outcomes to demonstrate benefit of systems thinking.
- Unfortunately, the authors are not aware of published data to address this question in stocker operations. Revised paper addresses in paragraph before conclusion. Authors also discuss need for applied research to measure both qualitative and quantitative benefits of systems thinking in stocker operations to address BRD.
- Links in the systems thinking framework across different disease manifestations encountered in the stocker operation also addressed in revised manuscript.
Thank you for the excellent points that helped to improve the article!
Reviewer 2 Report
Main question: Can systems thinking provide improved decision outcomes when addressing action options for stocker cattle health? - - - Yes, this is a relevant and interesting question.
This appears to be an original additional to the scientific literature - I have not seen similar articles addressing this question。
The authors' conclusions are consistent with the arguments presented - but this is the weakest part of the manuscript. The authors used the example of decisions about antimicrobial use to treat BRD to support systems thinking as an alternative method to linear thinking - but the contrast between linear thinking and systems thinking could be clearer. It may also be helpful to identify other decisions that stocker cattle operators/veterinarians make that could be improved with systems thinking.
The paper is very well-written and requires very little editorial/grammatical improvement.
Good job describing an important topic for veterinarians involved in the beef cattle supply chain. Lacks clearly defined action steps for a veterinarian working with stocker cattle clients - other than to implement systems thinking.
Author Response
- Conclusion: Revised conclusion more clearly contrasts linear thinking to systems thinking.
- Included suggestions to implement systems thinking as a decision-making tool across the structure of the operation.
Thank you for the excellent suggestions to improve the article!
Reviewer 3 Report
Dear authors,
The manuscript entitled: "Assessing the Role of Systems Thinking for Stocker Cattle Systems" submitted by CUMMINGS et al. to Veterinary Sciences, discussed the discipline of systems thinking as an alternative approach to address the complex nature of multifactorial disease manifestations, i.e., BRD. The core of your paper provides the framework to understand the essentials of systems thinking and apply critical fundamentals in a way that covers an aspect of absolute importance to animal health professionals and cattle breeders to control respiratory infections better. Also, it is written as an exciting and original piece of work, and I found it novel to be considered further for publication as the authors put much effort into solidifying their findings; however, I think that minor concerns need more clarification, as well I would suggest some hints for the manuscript improvement as follows:
1-To make the article more researchable, I recommend the title change as follows: Assessing the Role of Systems Thinking for Stocker Cattle Operations.
2- In the introduction, what did the authors mean by the word: thwart management.
3-In conclusion: it would be better if the authors could briefly put some suggestions to address the gaps in such vital protocols for future research applications of systems thinking in the animal production sector.
Author Response
- Excellent suggestion. Revised title
- "thwart" is intended to mean counteract or oppose efforts to improve the current condition, particularly when there is disagreement about the current condition of the "problem".
- Suggestion is included in final paragraph before conclusion. The authors recognize the need to address gaps and conduct applied research to measure the qualitative and quantitative benefits of implementing systems thinking as it relates to the BRD complex.
Thank you for the excellent suggestions that helped to improve the article!